# FDVLA: A Flow-Diffusion Vision-Language-Action Framework with Dual Reasoning Modulation

## Abstract

Recent advances in vision-language models (VLMs) have empowered robots to interpret natural language and perform complex manipulation tasks. Existing vision-language-action (VLA) frameworks typically adopt autoregressive decoding or diffusion-based strategies. While the former may lead to fragmented or less smooth trajectories, the latter often lacks explicit injection of reasoning semantics into the action generation process, which can affect the quality of generated actions. In this paper, we propose FDVLA, a unified framework integrating semantic reasoning with smooth and physically coherent action generation. We introduce a flow-diffusion mechanism that unifies global trajectory planning (via flow fields) and fine-grained action refinement (via diffusion) in a dual-headed policy, enabling physically coherent and stable action generation. Additionally, we design DualMod, a lightweight module that injects semantic signals into both velocity and noise prediction branches, thus integrating high-level reasoning into action generation. Extensive experiments across diverse simulated and real-world robotic tasks, demonstrate that FDVLA achieves solid performance, efficient inference, and shows robust generalization under a variety of task conditions.

## 1 Introduction

Recent advancements in vision-language models (VLMs) have significantly propelled the development of vision-language-action (VLA) systems, empowering robots to comprehend high-level instructions and execute grounded manipulation across a wide range of tasksKim et al. (2024); Bu et al. (2024); Liu et al. (2025a); Zheng et al. (2025); Zhu et al. (2024); Ze et al. (2024); Zhang et al. (2024); Pertsch et al. (2025); Brohan et al. (2022); Team (2024); Team et al. (2024).

Autoregressive VLA approaches such as SayCan, PaLM-E, VIMA, CaP, RT-2, and OpenVLA rely on token-based discretization of continuous actions to leverage the semantic reasoning capabilities of large language models (LLMs) Ahn et al. (2022); Driess et al. (2023); Jiang et al. (2022); Liang et al. (2022); Zitkovich et al. (2023); Wang et al. (2023). Despite their strong reasoning abilities, these autoregressive methods inherently disrupt trajectory continuity due to discretization, leading to jerky and less precise movements in practical robotic execution. Moreover, the inherent computational overhead in token-based prediction makes inference inefficient, restricting their application in real-time robotic control. Meanwhile, some VLA approaches Jiang et al. (2022); Wang et al. (2023); Cheang et al. (2024); Huang et al. (2024); Li et al. (2023b); Liu et al. (2024b); Wu et al. (2023) incorporate an MLP- or LSTM-based policy head that transforms LLM output embeddings into continuous action poses, enabling direct regression of actions. However, these regressive methods lack the flexibility to model multi-modal action distributions present in real-world tasks.

To address the above limitations, recent diffusion-based VLA methods Wen et al. (2025b); Liu et al. (2025b); Chen et al. (2024b) incorporate denoising diffusion processes, and flow-based models such as $\pi_0$ Black et al. learn continuous vector fields for trajectory guidance. While these methods produce smoother and more precise trajectories, most existing approaches operate independently of the semantic reasoning process. Typically conditioned only on static embeddings from pretrained VLMs, these diffusion components lack dynamic semantic adaptability, thus underutilizing the reasoning capabilities of the underlying foundation models Wen et al. (2025a). Furthermore, these

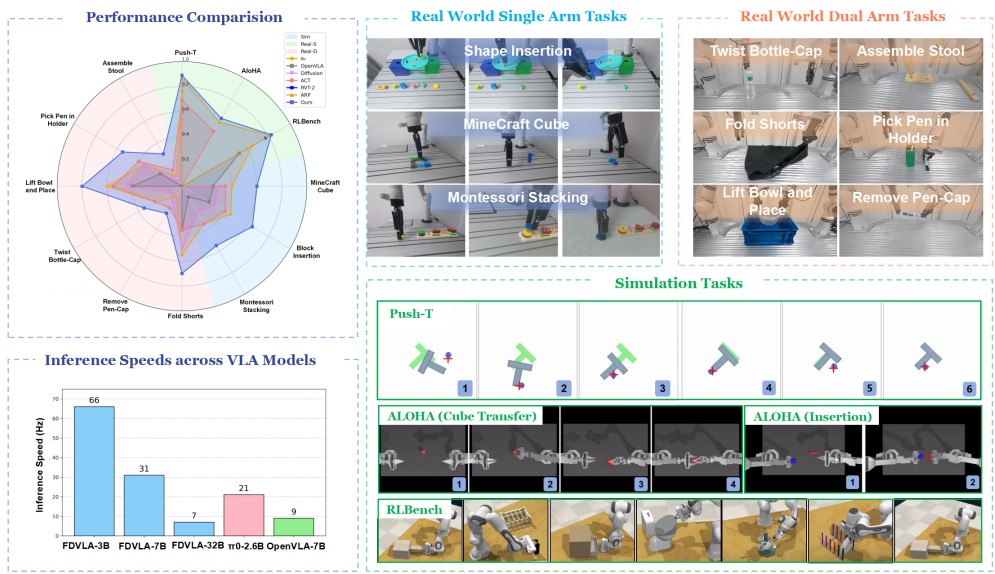

Figure 1: Overview of FDVLA's performance and task coverage. FDVLA is evaluated across diverse simulation and real-world settings, including single- and dual-arm tasks. Many tasks are custom-designed to span a wide range of manipulation challenges, showcasing strong performance and generalization across varied scenarios and environmental conditions.

diffusion-based models usually omit explicit physical constraints (e.g., velocity fields), limiting trajectory precision and smoothness, particularly in challenging manipulation tasks. Motivated by these limitations, we raise a critical question: *How can we design a unified vision-language-action framework that explicitly integrates semantic reasoning, physically consistent motion generation, and multi-modal probabilistic representations within a coherent modeling structure?*

In this paper, we present FDVLA (Flow-Diffusion Vision-Language-Action), a unified framework that addresses the above challenges by combining flow-based velocity modeling with diffusion-based denoising to generate smooth and physically consistent trajectories. To seamlessly integrate semantic reasoning into the action generation process, we introduce DualMod, a lightweight modulation module that dynamically injects instruction semantics into both the flow and denoising stages. Our architecture is empirically validated across a wide range of simulated and real-world manipulation tasks, including single- and dual-arm robots operating in varied spatial layouts, object types, and lighting conditions (see Fig. 1). To summarize, our contributions are three-fold:

- We propose FDVLA, a unified vision-language-action framework that seamlessly integrates **flow-diffusion**, our novel action generation mechanism based on flow fields and denoising dynamics, with the reasoning capabilities of large language models (LLMs). This design enables generated action that is both smooth and semantically informed by linguistic instructions.

- Furthermore, we introduce DualMod, a lightweight reasoning-guided modulation module that softly injects instruction semantics into both the flow field and denoising dynamics. This innovation allows reasoning signals to modulate action generation, providing a foundation for interpretable and adaptable robotic behaviors across different scenarios.

- We evaluate FDVLA across a wide range of simulated (Push-T, Aloha, RLBench) and real-world tasks, including shape sorting, magnetic cube assembly, chair assembly, and Montessori-style geometric stacking, and observe robust performance and generalization to novel and cognitively demanding task settings.

## 2    RELATED WORK

**Vision-Language-Action Models.**    Recent advances in VLMs Alayrac et al. (2022); Bai et al. (2023); Gao et al. (2023); Li et al. (2023a); Liu et al. (2023c) have significantly expanded the capabilities of robotic systems, enabling them to interpret complex instructions and perform semantically aligned manipulation tasks. A series of vision-language-action (VLA) frameworks have emerged to leverage such reasoning capabilities. Early approaches rely on token-based autoregressive action generation Brohan et al. (2023a); Wang et al. (2024b), enabling language-conditioned control through next-token prediction. However, this often requires discretizing continuous action trajectories, which can introduce temporal discontinuities and reduce execution smoothness. Regression-based methods Zhu et al. (2023) mitigate this by directly predicting continuous actions via MLP-based or LSTM-based heads, but they typically lack probabilistic expressivity and struggle to scale across diverse tasks and scenes. To overcome these limitations, diffusion-based policies have recently been introduced to VLA systems to generate smoother and continuous action Wen et al. (2024); Li et al. (2024); Chen et al. (2024b). Despite their potential, current integration of diffusion model remains loosely coupled with the VLM backbone, limiting their ability to fully leverage semantic reasoning during policy generation.

**Diffusion Policies in VLA Models.**    Diffusion models Ho et al. (2020); Ramesh et al. (2022); Chen et al. (2024a); Ma et al. (2024); Xing et al. (2024) have achieved state-of-the-art performance in image and video generation tasks Ho et al. (2020); Ramesh et al. (2022), and are increasingly adopted in robotics to model complex, multimodal action distributions. Early works such as Diffusion Policy Chi et al. (2023) demonstrated the effectiveness of denoising-based sampling in imitation learning, inspiring subsequent extensions to domains such as 3D grasping Ke et al. (2024) and transformer-based policy generation Team et al. (2024); Liu et al. (2024c). To integrate diffusion models with large pretrained VLMs, recent methods incorporate diffusion heads into VLA frameworks. For instance, $\pi_0$ utilizes a flow-matching head, which is a vector-field-based approach closely related yet distinct from standard denoising diffusion, for trajectory generation. TinyVLA Wen et al. (2025a) introduces a diffusion head following a lightweight VLM, while methods Chen et al. (2024b); Wen et al. (2025b) decouple semantic reasoning and action prediction into separate VLM and diffusion modules. However, most existing designs treat the diffusion policy as an isolated module that conditions only on static embeddings extracted by the VLM, thereby limiting the utilization of the semantic reasoning capabilities inherent in foundation models. In contrast, our work proposes a unified architecture that closely integrates semantic reasoning and trajectory generation through a novel flow-diffusion mechanism, enabling smooth and physically consistent actions. Moreover, we introduce a lightweight DualMod module that dynamically modulates the denoising process using high-level semantic features extracted from the VLM.

## 3    METHOD

Our goal is to build a unified vision-language-action framework that generates continuous actions while leveraging large language models to inject semantic reasoning into the process. Developing such an integrated framework raises several challenges, including: (i) designing an architecture that seamlessly integrates flow matching and diffusion-based denoising within a single policy; and (ii) enabling dynamic semantic reasoning signals to enhance action generation, while maintaining inference efficiency and scalability. In the following subsections, we first present an overview of the FDVLA architecture (Section 3.1), describing how vision, language, and state information are encoded and processed by the pretrained VLM. Section 3.2 then details the flow-diffusion formulation, including how actions are represented as velocity fields and iteratively refined via conditional denoising. Finally, Section 3.3 introduces our DualMod module, which bridges high-level semantic reasoning and low-level action synthesis.

### 3.1    OVERVIEW

Our FDVLA framework integrates visual observations, language instructions, and robot state embeddings into a unified token sequence for multimodal reasoning and action generation. As shown in Figure 2, the system comprises a powerful visual encoder, a pretrained vision-language model, and two key modules: FlowDiffusion for action generation and DualMod for semantic modulation.

**Visual Encoder.**    We adopt **SigLIP** Zhai et al. (2023) as the vision encoder to extract dense features from multi-view RGB observations. Each view is independently encoded, and the resulting tokens

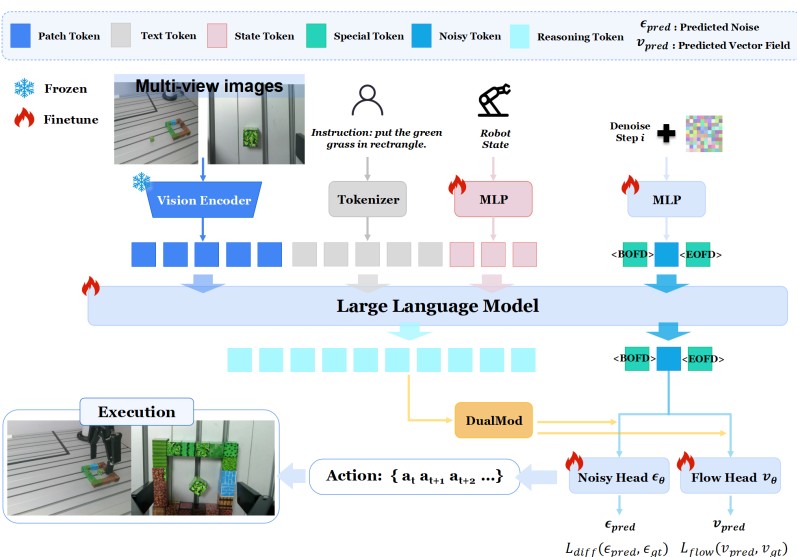

Figure 2: Overview of our proposed FDVLA framework. FDVLA is a unified vision-language-action system that builds upon a pretrained Vision-Language Model (VLM), introducing a flow-based diffusion module for robot action generation. The input comprises multi-view visual observations, natural language task prompts, and robot state embeddings, which are projected into the token space. We insert a noisy action token segment (delimited by `<BOFD>` and `<EOFD>`), which encodes both the denoising timestep and the noisy action embedding for each iteration. This segment provides necessary context for both the flow head and the denoising head during decoding.

are concatenated to form a unified visual representation. This design captures complementary spatial cues across viewpoints without explicit cross-view fusion. The final concatenated features are projected into the VLM token space.

**Vision-Language Backbone.** We use the publicly available **Qwen2.5-VL** model Bai et al. (2025); Wang et al. (2024a); Bai et al. (2023) as the VLM backbone, exploring three model sizes: **3B**, **7B**, and **32B**. The VLM jointly processes tokenized language instructions and projected visual features, enabling semantic grounding across both observations and instructions.. All VLM parameters are initialized from released checkpoints. During training, we freeze the vision encoder and fine-tune the VLM for task adaptation. The overall architecture is flexible and modular, and the VLM backbone can be replaced with other powerful pretrained models (e.g., LLaVA Liu et al. (2023b;a; 2024a), GPT-4o OpenAI et al. (2024). In the next section, we show how FlowDiffusion and DualMod are designed for action generation and reasoning modulation.

### 3.2 FLOWDIFFUSION

Flow-based models excel at producing smooth and physically consistent trajectories through explicit velocity field modeling. While diffusion-based policies enable expressive action generation and can capture complex distributions, they may require iterative sampling and sometimes exhibit high-frequency jitter. To harness the strengths of both approaches, we propose **FlowDiffusion**, a unified action policy that integrates flow matching and diffusion denoising within a dual-headed policy architecture.

Given a noisy action input $A_t^\tau$ at timestep $t$, FlowDiffusion predicts: (1) a velocity field $v_\theta(A_t^\tau, o_t)$ representing the global direction of the target trajectory, and (2) a residual noise $\epsilon_\theta(A_t^\tau, \tau, o_t)$ capturing fine-grained corrections from the diffusion process. The two heads are jointly optimized under a composite objective:

$$\mathcal{L} = \underbrace{\|\hat{\epsilon}_\theta - z\|^2}_{\text{Denoising}} + \lambda_1 \underbrace{\left\|v_\theta - \frac{A_0 - A_t}{T - t}\right\|^2}_{\text{Flow Matching}} + \lambda_2 \underbrace{\|\nabla_{A_t}\hat{\epsilon}_\theta - v_\theta\|^2}_{\text{Flow Consistency}} \quad (1)$$

The first term corresponds to the standard DDPM loss. The second term encourages the velocity prediction to align with the true trajectory direction, while the third term enforces local-global consistency by aligning the gradient of denoising with the predicted velocity field. The theoretical motivation and a more detailed discussion are provided in Appendix B.

For inference, we adopt a DDIM-style few-step sampling process, guided by the following forward integration rule:

$$A_t^{\tau+\delta} = A_t^\tau + \delta \left( v_\theta(A_t^\tau, o_t) + \alpha \epsilon_\theta(A_t^\tau, \tau, o_t) \right) \tag{2}$$

Here, $\delta$ denotes the integration step size, and $\alpha$ controls the influence of residual refinement. This unifies coarse motion planning and fine-scale correction, enabling FlowDiffusion to achieve stable training, few-step inference, and controllable multimodal generation. The effectiveness of Flow-Diffusion is further validated in Section 4.4.

### 3.3 DualMod: Joint Modulation of Flow and Denoising for Fine-Grained Reasoning

We introduce **DualMod**, a lightweight yet effective reasoning modulation module that enhances both the flow prediction and the denoising process via a shared reasoning vector. Rather than relying on recursive rollouts or autoregressive decoding, our method integrates reasoning information directly into the generation process, offering structured semantic control without additional architectural burdens.

Inspired by recent work(FiLM) Birnbaum et al. (2019); Wen et al. (2024); Brohan et al. (2022); Shi et al. (2024) based on Feature-wise Linear Modulation (FiLM) Perez et al. (2018), DualMod injects task-conditioned semantic control into both generation branches.

Formally, given a language-grounded reasoning vector $r \in \mathbb{R}^d$ (extracted via global pooling from the final layer of VLM), we generate modulation weights $(\gamma_v, \beta_v)$ and $(\gamma_\epsilon, \beta_\epsilon)$ through MLPs:

$$\gamma_v, \beta_v = \text{MLP}_v(r), \quad \gamma_\epsilon, \beta_\epsilon = \text{MLP}_\epsilon(r). \tag{3}$$

These weights are used to modulate the respective branch features $(h_v, h_\epsilon)$ before head prediction:

$$\tilde{h}_v = \gamma_v \cdot h_v + \beta_v, \quad \tilde{h}_\epsilon = \gamma_\epsilon \cdot h_\epsilon + \beta_\epsilon. \tag{4}$$

In this way, the reasoning vector jointly guides both the velocity and residual pathways, ensuring alignment between coarse motion intent and fine-grained corrective signals. This cross-branch semantic coordination allows the policy to dynamically adjust trajectory patterns based on high-level task semantics, improving generalization under ambiguous or multi-modal prompts. Moreover, DualMod introduces negligible computation overhead and is fully differentiable, making it compatible with end-to-end training alongside the VLM backbone used in our FDVLA architecture. The effectiveness of DualMod's joint modulation design is further validated by ablation experiments (see Section 4.5), which show that removing semantic modulation from either branch leads to noticeable performance degradation.

## 4 Experiment

In this section, we evaluate the effectiveness of FDVLA for embodied control across both simulation and real-world settings. In Section 4.2, we benchmark FDVLA against several state-of-the-art baselines on a range of simulation tasks, covering diverse control frequencies and action spaces. In Section 4.3, we examine FDVLA's performance on real-world single-arm and dual-arm tasks, demonstrating its ability to generalize across embodiments and manipulation types. Section 4.4 analyzes the core flow-diffusion mechanism and its impact on trajectory quality. Section 4.5 presents an ablation study of the DualMod module to assess the contribution of reasoning-aware modulation. Finally, Section 4.6 explores model scaling behavior using a toy sorting task involving both seen and unseen objects.

### 4.1 Experimental Setting

To assess its effectiveness, we evaluate FDVLA across both simulation and real-world environments. Simulation tasks include Push-T, ALOHA, and RLBench, covering a range of position control, dexterous manipulation, and long-horizon planning scenarios. For real-world experiments, we design a

diverse set of single-arm and dual-arm manipulation tasks, such as shape insertion, magnetic cube assembly, Montessori geometric stacking, wood chair assembly, folding shorts, pick bowl and placement, pen insertion, and pen cap removal. These tasks test spatial reasoning, bimanual coordination, and the ability to handle novel or complex objects.

**Implementation Details** FDVLA is pretrained on large-scale datasets Droid and OXE of robot manipulation demonstrations and then fine-tuned on both simulated and self-collected real-world datasets. The model uses a Qwen2.5-VL Bai et al. (2025) backbone in three parameter sizes (3B, 7B, and 32B), with a frozen visual encoder during fine-tuning. For efficient adaptation, we employ LoRA on the vision-language backbone. All training runs are performed on NVIDIA A800 GPUs. Simulation datasets include 100 trajectories per single-arm task and 150 trajectories per dual-arm task. Single-arm experiments use a Kinova Gen3 robot, and dual-arm experiments are performed on a RM65-B dual-arm platform. Real-world data collection employs both wrist-mounted RGB cameras and external RealSense 435 cameras (using only the RGB stream) to provide diverse visual perspectives. During fine-tuning, related tasks and embodiments are grouped to support cross-task generalization.

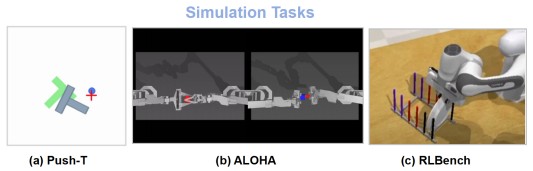

Figure 3: Overview of the simulation environments. (Details can be found in Appendix A.1.)

## 4.2 SIMULATION EXPERIMENTS

We evaluate FDVLA on a range of simulation benchmarks, including Push-T Chi et al. (2023), ALOHA Zhu et al. (2023), and RLBench James et al. (2020). Normalized task success rates are reported in Table 1. For each method, results on Push-T and RLBench are averaged over three independent runs, while ALOHA results are averaged over five runs to ensure statistical reliability. Across all tested tasks, FDVLA consistently outperforms strong baselines such as $\pi_0$ Huang et al. (2023), OpenVLA Wang et al. (2024b), and ARP Zhang et al. (2025), achieving higher success rates and improved planning efficiency.

Table 1: Success rates on simulation benchmarks. Notably, FDVLA demonstrates robust performance on both long-horizon tasks (RLBench) and dexterous multi-step manipulations (ALoHA), highlighting its versatility and effectiveness in diverse simulated environments.

| Task | Ours(3B) | $\pi_0$ | OpenVLA | Diffusion | ACT | RVT-2 | ARP |
|---|---|---|---|---|---|---|---|
| Push-T | **0.89** | 0.763 | 0.597 | 0.788 | 0.69 | / | 0.876 |
| ALoHA | **0.629** | / | / | / | 0.508 | / | 0.595 |
| RLBench | **0.826** | 0.65 | 0.53 | / | / | 0.772 | 0.813 |

Table 2: Quantitative results across real-world tasks. Real-S and Real-D refer to Single-Arm and Dual-Arm Real-World settings. Metrics are task success rates (normalized to [0, 1]).

| Task Type | Task Name | Models | | | | | | |
|---|---|---|---|---|---|---|---|---|
| | | **Ours** | $\pi_0$ | OpenVLA | Diffusion Policy | ACT | RVT-2 | ARP |
| Real-S | Minecraft Cube | **0.60** | 0.40 | 0.25 | 0.30 | 0.35 | / | / |
| | Block Insertion | **0.65** | 0.45 | 0.25 | 0.30 | 0.40 | / | / |
| | Montessori Geometric Stacking | **0.55** | 0.35 | 0.10 | 0.25 | 0.35 | / | / |
| Real-D | Fold Shorts | **0.70** | 0.55 | 0.35 | 0.50 | 0.35 | / | / |
| | Remove Pen-Cap | **0.25** | 0.10 | 0.00 | 0.15 | 0.20 | / | / |
| | Twist Bottle Cap | **0.35** | 0.25 | 0.00 | 0.10 | 0.25 | / | / |
| | Lift Bowl and Place | **0.80** | 0.60 | 0.40 | 0.45 | 0.55 | / | / |
| | Pen in Holder | **0.55** | 0.35 | 0.20 | 0.35 | 0.40 | / | / |
| | Assemble Stool | **0.30** | 0.10 | 0.00 | 0.05 | 0.15 | / | / |

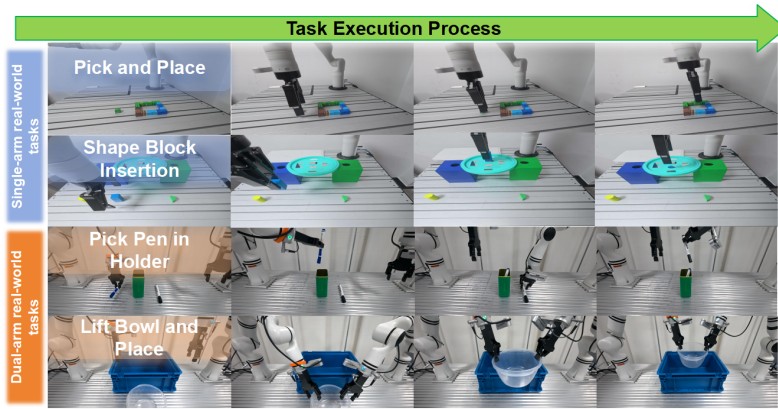

Figure 4: Task execution process of FDVLA on single-arm and dual-arm real-world manipulation tasks. Each row illustrates the key frames along the execution process for a specific task. More visualizations are provided in Appendix.A.2.

### 4.3 REAL-WORLD EXPERIMENTS

To thoroughly assess real-world applicability, we deploy FDVLA on a variety of challenging single-arm and dual-arm robotic manipulation tasks. These experiments cover a broad spectrum of object types, manipulation skills, and environmental conditions, enabling an in-depth evaluation of the model's robustness and generalization in practical settings. The results are summarized as follows.

**Quantitative Results.**    Table 2 presents the success rates for a range of real-world manipulation tasks. On single-arm tasks, FDVLA achieves a success rate of 60% for Minecraft Cube assembly, 65% for Shape Insertion, and 55% for Montessori Stacking. These results consistently exceed those of the baseline models. In dual-arm settings, FDVLA also delivers strong results. The model achieves 80% success on Lift Hat and Place, 70% on Fold Shorts, and 55% on Pen in Holder. These outcomes highlight FDVLA's ability to coordinate both arms. The robust performance across a variety of tasks suggests that FDVLA is effective at both spatial reasoning and coordinated action generation. Overall, these quantitative results indicate that FDVLA not only generalizes well to new tasks and environments but also maintains stable action execution and reliable planning across diverse manipulation scenarios.

**Qualitative Results.**    Figure 4 shows representative real-world manipulation sequences. On single-arm tasks, FDVLA smoothly handles multi-step cube placement (Minecraft Cube), accurately inserts objects into tight-fitting slots (Shape Insertion), and reliably stacks components in structured sequences (Montessori Stacking). In dual-arm scenarios such as Lift Hat and Place, the robot achieves coordinated motion without explicit jitter. Overall, qualitative outcomes demonstrate FDVLA's solid spatial reasoning and stable action execution, highlighting its adaptability and robustness in practical environments. Additional examples, including typical failure modes, are provided in supplementary materials. Overall, these real-world results strongly validate FDVLA's practical applicability, robustness, and superior generalization capabilities in complex and dynamic manipulation scenarios. Additional failure case analyses are provided in Appendix D. We further evaluate the robustness of FDVLA under environmental shifts. Generalization results under unseen object, position, background, and lighting conditions are provided in Appendix F.

### 4.4 FLOW-DIFFUSION ANALYSIS

To further analyze the impact of modeling strategy on robotic action generation, we compare three representative policies: (1) **Flow-only**, which predicts velocity fields for direct ODE-based action integration; (2) **Diffusion-only**, which relies solely on iterative denoising without explicit planning guidance; and (3) **FDVLA (Ours)**, which unifies flow-guided planning and diffusion-based correction. Table 3 summarizes the quantitative results on representative tasks spanning position control,

Table 3: Comparison of Flow-only, Diffusion-only, and FDVLA on representative manipulation tasks. Smoothness is measured by jerk, defined as the average second-order difference of the trajectory, where lower values indicate smoother motion.

| Task | Flow-only | Diffusion-only | FDVLA-3B | Smooth. (F) | Smooth. (D) | Smooth. (Ours) |
|------|-----------|----------------|----------|-------------|-------------|----------------|
| Push-T | 70% | 80% | **89%** | 0.210 | 0.233 | **0.199** |
| Fold Shorts | 55% | 60% | **70%** | 0.272 | 0.300 | **0.264** |
| Lift Bowl and Place | 65% | 70% | **80%** | 0.267 | 0.289 | **0.257** |

dexterous manipulation, and sequential assembly. FDVLA consistently achieves higher task success rates and produces smoother action trajectories (as measured by jerk) compared to Flow-only and Diffusion-only baselines. Flow-only methods yield physically smooth but less precise actions. Diffusion-only models achieve higher task success rates than flow-only baselines in some cases, but often at the cost of increased trajectory variability (as reflected in the smoothness metric). FDVLA consistently delivers both high task success and smooth trajectory execution across all evaluated tasks. These results validate the benefit of jointly modeling velocity fields and denoising processes for robust and precise robotic control.

## 4.5 DualMod Ablation Study

To comprehensively evaluate the contribution of our DualMod module within the FDVLA framework, we conducted targeted ablation experiments. DualMod functions by injecting semantic reasoning signals into both the velocity prediction and noise prediction process. To isolate the impact of this component, we compared the full FDVLA model (with DualMod) to three ablated variants: one with DualMod removed entirely, one with modulation applied only to the denoising branch, and one with modulation applied only to the flow branch. Table 4 summarizes the quantitative results for these variants on selected tasks from our simulation and real-world benchmarks. Our findings clearly demonstrate that the full DualMod implementation consistently achieves the highest success rates across all evaluated tasks. Removing DualMod entirely leads to a significant performance degradation, highlighting its crucial role in achieving robust semantic alignment and physically coherent action generation. Furthermore, selectively disabling modulation in either the flow or denoising branches results in intermediate performance drops, suggesting that DualMod's joint modulation design effectively coordinates semantic reasoning with both coarse-grained trajectory guidance and fine-grained trajectory refinement. These ablation results empirically validate the effectiveness and necessity of DualMod within our FDVLA framework, underscoring its contribution to robust and adaptable robotic performance.

Table 4: Ablation results for DualMod modulation strategies (normalized success rates, evaluated over 20 rollouts per task).

| Task | FDVLA(3B) | w/o DualMod | w/o Flow Mod. | w/o Denoising Mod. |
|------|-----------|-------------|---------------|--------------------|
| Block Insertion | 0.65 | 0.55 | 0.60 | 0.60 |
| Fold Shorts | 0.70 | 0.60 | 0.65 | 0.65 |
| Lift Bowl and Place | 0.80 | 0.70 | 0.75 | 0.75 |

These ablation results empirically validate the effectiveness and necessity of DualMod within our FDVLA framework, underscoring its contribution to robust and adaptable robotic behaviors.

## 4.6 Model Scaling Analysis

We evaluated FDVLA models of different sizes on a toy sorting task. Each model, with 3B, 7B, or 32B parameters, was tested on sorting toy dolls and toy cars. We used two settings: in-distribution, where all toys were seen during training, and out-of-distribution, which introduced unseen toys. For each setting, we did five rollouts with 20 objects per rollout, so there were 100 decisions per model. The results are shown in Table 5. As the model size increased, accuracy improved in both settings. The 3B model reached 65% accuracy on familiar toys and 37% on new toys. The 7B model did better, and the 32B model performed best, with 80% and 63% accuracy. These results show that scaling up FDVLA leads to more reliable performance, especially when the robot faces objects it has not seen before.

Table 5: Performance of FDVLA models of different parameter sizes on the toy sorting task. Accuracy is reported as average percentage of correctly sorted objects out of 100 per setting.

|                     | FDVLA-3B | FDVLA-7B | FDVLA-32B |
|---------------------|----------|----------|-----------|
| In-Distribution     | 65%      | 71%      | 80%       |
| Out-of-Distribution | 37%      | 45%      | 63%       |

Larger models show clear gains in both memorizing what they have seen and handling new cases. The biggest model, FDVLA-32B, was the most robust when sorting unfamiliar toys. This suggests that a larger model helps the robot generalize better in open-ended scenarios.

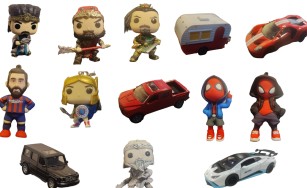

Figure 5: Sample toy dolls and vehicles from the sorting task.

### 4.7 EFFICIENT INFERENCE

Fast inference is critical for deploying VLA models in real-world settings, where timely control is required for closed-loop robotic applications. As model scale increases, maintaining high inference speed becomes increasingly challenging. To address this, we evaluate FDVLA models of different sizes on a single NVIDIA A800 GPU, leveraging the vLLMKwon et al. (2023) framework for efficient serving. Table 6 reports the control frequencies achieved by FDVLA at three parameter scales, alongside baseline models of similar size. Notably, FDVLA-3B achieves a control frequency of 66 Hz, which is substantially faster than OpenVLA-7B (9 Hz) and $\pi_0$-2.6B (21 Hz). Even for the larger FDVLA-7B and FDVLA-32B variants, inference remains efficient at 31 Hz and 7 Hz, respectively. These results highlight the scalability and practical applicability of our approach for real-time robot control. While inference acceleration frameworks such as vLLM can further improve throughput, we observe that FDVLA maintains efficient inference even without specialized quantization or architectural modifications. However, as with other VLA modelsLiu et al. (2025b); Wen et al. (2025b), aggressive quantization (e.g., 8-bit or 4-bit) can introduce non-trivial accuracy drops, suggesting that future work on quantization-aware VLA design may be required for optimal deployment.

Table 6: Inference speeds (Hz) of VLA models on a single NVIDIA A800 GPU.

| Model      | FDVLA-3B | FDVLA-7B | FDVLA-32B | OpenVLA-7B |
|------------|----------|----------|-----------|------------|
| Speed (Hz) | 66       | 31       | 7         | 9          |

## 5 CONCLUSION

We introduce FDVLA, a vision-language-action framework that unifies flow matching and diffusion-based refinement for continuous trajectory generation while tightly integrating semantic reasoning via DualMod. FDVLA further integrates DualMod, a lightweight modulation module that injects reasoning signals from large language models into both noise prediction and flow estimation, to enhance action generation. This design enables robotic actions that are semantically grounded, physically coherent, and temporally smooth. In extensive simulation and real-world single-arm and dual-arm manipulation benchmarks, FDVLA achieves consistently strong success rates, produces smooth and coherent trajectories, and retains efficient inference. Our findings suggest that coupling flow-diffusion with reasoning-aware modulation offers a promising direction for scalable and robust VLA modeling.

## ETHICS STATEMENT

Our work develops FDVLA, a vision-language-action framework designed for robotic manipulation tasks in simulated and real-world settings. All experiments were conducted using either publicly available simulation benchmarks (e.g., Push-T, ALOHA, RLBench) or real-world tasks in controlled lab environments with inanimate objects. No human subjects, sensitive personal data, or animal experiments were involved. The visual data used for training and evaluation were either self-collected in lab settings or sourced from open-access datasets. We ensured that no copyrighted or personally identifiable materials were used.

We believe FDVLA poses minimal ethical risks. However, as with all vision-language robotic systems, there exists potential for misuse in surveillance or unsafe deployment. To mitigate this, we publish this work solely for academic research and prohibit its use in high-risk domains without proper safety and ethical safeguards. We encourage the community to explore safety-aligned training, environment simulation fidelity, and robust policy evaluation as future directions to ensure responsible deployment of such systems.

## REPRODUCIBILITY STATEMENT

To support reproducibility, we will release the full codebase of FDVLA, including model implementation, training pipelines, and inference scripts. All simulation environments used in this paper (Push-T, ALOHA, RLBench) are open-source and publicly available. We provide detailed descriptions of the training settings, hyperparameters, model sizes, and evaluation metrics in the main paper and Appendix. For real-world experiments, we include all task definitions, robot configurations, and data collection protocols to ensure others can replicate our results. The pretrained VLM backbone (Qwen2.5-VL) is publicly released by the original authors. Dataset links, LoRA configurations, and real-world rollout videos will be provided upon acceptance. We will also publish ablation and failure case data to aid further verification.

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

Table 7: Quantitative results across simulation and real-world tasks. Real-S and Real-D refer to Single-Arm and Dual-Arm Real-World settings. Metrics are task success rates.

| Task Type / Model | Task Name | $\pi_0$ | OpenVLA | Diffusion Policy | ACT | RVT-2 | ARP | Ours |
|---|---|---|---|---|---|---|---|---|
| Sim | Push-T | 76.3 | 59.7 | 78.8 | 69 | / | 87.1 | **89** |
| | ALoHA | / | / | / | 50.8 | / | 59.45 | **62.85** |
| | RLBench | 65 | 53 | / | / | 77.2 | 81.3 | **82.6** |
| Real-S | Minecraft Cube | 0.40 | 0.25 | 0.30 | 0.35 | / | / | **0.60** |
| | Block Insertion | 0.45 | 0.25 | 0.30 | 0.40 | / | / | **0.65** |
| | Montessori Geometric Stacking | 0.35 | 0.10 | 0.25 | 0.35 | / | / | **0.55** |
| Real-D | Fold Shorts | 0.55 | 0.35 | 0.50 | 0.35 | / | / | **0.70** |
| | Remove Pen-Cap | 0.10 | 0.00 | 0.15 | 0.20 | / | / | **0.25** |
| | Twist Bottle Cap | 0.25 | 0.00 | 0.10 | 0.25 | / | / | **0.35** |
| | Lift Bowl and Place | 0.60 | 0.40 | 0.45 | 0.55 | / | / | **0.80** |
| | Pen in Holder | 0.35 | 0.20 | 0.35 | 0.40 | / | / | **0.55** |
| | Assemble Stool | 0.10 | 0.00 | 0.05 | 0.15 | / | / | **0.30** |

# A   DETAILS ABOUT SIM AND REAL TASKS

## A.1   SIMULATION ENVIRONMENTAL DETAILS

We evaluate FDVLA on three simulation environments: Push-T, ALOHA, and RLBench. Each environment brings its own set of challenges and helps us assess the generalization and reasoning abilities of our method.

**Push-T**: The robot must push a T-shaped object to overlap a target region on the table. This task requires accurate spatial reasoning and multi-step planning. The action space is a 2D pointer, and the solution space is highly multimodal. The robot must handle long-horizon trajectories and adapt to various object positions.

**ALOHA**: We use two representative tasks—Cube Transfer and Cube Insertion. In Cube Transfer, the robot picks up a cube and moves it to a designated location using both arms. In Cube Insertion, the robot needs to insert a block into a tight slot. Both tasks involve 14-joint position control for bimanual arms. The high action dimensionality and short execution horizon make these tasks especially challenging for coordination and precision.

**RLBench**: This environment offers a diverse set of language-conditioned tasks. Each task requires the robot to manipulate objects such as blocks, drawers, or cups, using a 6DoF end-effector pose and discrete gripper actions. The tasks cover a broad range of skills, including pick-and-place, stacking, insertion, and open/close operations. RLBench scenarios test both general spatial reasoning and the model's ability to follow language instructions.

Together, these settings provide a thorough benchmark for evaluating vision-language-action models in simulation.

## A.2   ADDITIONAL REAL-WORLD TASK DEMONSTRATION

More visualization of FDVLA in both single and dual arms setting. As shown in Figure 6.

## A.3   REAL-WORLD TASK DETAILS

**Single-Arm Tasks.**   We tested the performance of our model on tasks requiring precise spatial reasoning and manipulation:

- **Shape Insertion**: Inserting geometric objects into matching holes to evaluate spatial recognition.

- **MineCraft Cube Assembly**: Assembling magnetic cubes into specific structures, requiring spatial planning and contact reasoning.

- **Montessori Stacking**: Stacking rings onto pegs by size, assessing fine-grained spatial reasoning.

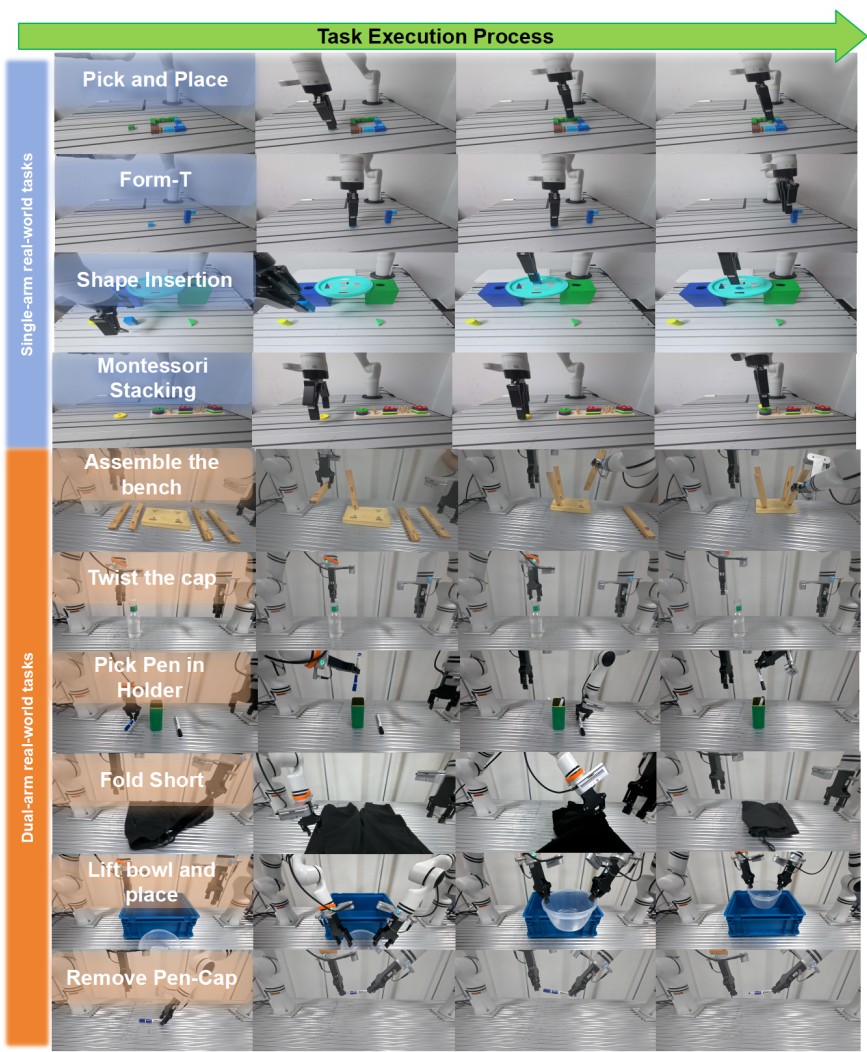

Figure 6: Additional visualizations of FDVLA performing diverse real-world manipulation tasks in both single-arm and dual-arm settings. The figure includes representative tasks such as pick and place, object insertion, stacking, folding, and assembly. These results highlight FDVLA's ability to generalize across varied task categories and coordination requirements.

**Dual-Arm Tasks.** To assess coordination between two robotic arms, we designed complex manipulation scenarios:

- **Fold Shorts**: Collaborative folding of cloth items, demanding coordinated movements.
- **Remove Pen-Cap**: Removing pen caps using precise bimanual coordination.
- **Twist Bottle Cap**: Twisting open bottle caps, evaluating dexterous manipulation skills.
- **Lift Bowl and Place**: Picking and placing hats horizontally and stably using both arms.
- **Pen in Holder**: Accurately inserting pens into holders with serial left and right arms precision.
- **Assemble Stool**: Constructing wooden stools, involving sequential assembly reasoning and manipulation.

All demonstrations were collected using master-puppet teleoperation with a dual-arm robotic platform for bimanual tasks and a Franka Panda robot for single-arm tasks.

Table 8: VQA for FDVLA. We test FDVLA's ability to answer questions based on visual signals.

| Question | Object/Scene | Answer | Y/N |
|---|---|---|---|
| What's the object? | | SUV Car | 51 |
| | | Toy Cartoon | 51 |
| | | Keys | ✓ |
| What's the color? | | White | 51 |
| | | Green | 51 |
| | | Silver | 51 |
| Describe the scene. | | A blue cube is stacked on top of a green rectangular block. | 51 |
| | | A Batman figurine stands next to a black SUV. | 55 |

## B  FDVLA FLOW CONSISTENCY LOSS: THEORETICAL MOTIVATION

**Theoretical Justification for Flow Consistency Loss.** We follow the theoretical insight from score matching and Tweedie's formula Ho et al. (2020); Song et al. (2020), which connect the gradient of the optimal denoiser in diffusion models with the underlying probability flow.

Let $A_t$ be the noisy action at time $t$ generated by the diffusion process, and let $\epsilon_\theta(A_t, t)$ be the optimal denoising function learned by the model. According to Tweedie's formula, the conditional mean of the clean data $A_0$ given $A_t$ can be expressed as:

$$\mathbb{E}[A_0|A_t] = A_t + \sigma_t^2 \mathbb{E}[\nabla_{A_t} \log p(A_t)]. \tag{5}$$

In the setting of DDPMs, the gradient of the optimal denoiser (with respect to $A_t$) approximates the probability flow (Stein score):

$$-\nabla_{A_t} \epsilon_\theta(A_t, t) \approx \frac{A_0 - A_t}{T - t}, \tag{6}$$

where $\frac{A_0 - A_t}{T-t}$ is the ground-truth velocity field in flow matching.

Thus, enforcing

$$\nabla_{A_t} \hat{\epsilon}_\theta \approx v_\theta, \tag{7}$$

where $v_\theta$ is the vector field predicted by the flow head, aligns the local corrections (from denoising) with the global trajectory direction (from flow).

This provides a self-consistency target for the flow head and regularizes the denoiser to respect the overall probability flow, leading to more stable and physically plausible action generation.

For further theoretical details, see Ho et al. (2020); Song et al. (2020).

## C  VISUAL QUESTION ANSWERING ABILITY OF FDVLA

Recent works have suggested that pretraining on vision-language data can preserve basic conversational and visual reasoning abilities in VLA models Brohan et al. (2023b); Wen et al. (2025b). Although FDVLA is not explicitly co-trained on vision-language datasets, we find that it retains a certain level of visual question answering (VQA) capability, owing to the use of a powerful pretrained VLM backbone. Table 8 presents several representative examples.

FDVLA can accurately answer questions about object identity, color, and spatial relations in some cases. For example, the model reliably identifies common objects and their colors, but occasionally fails to distinguish between similar objects or recognize unfamiliar items, especially when they differ from its pretraining data. In scene description tasks, FDVLA demonstrates an ability to interpret spatial relationships but may struggle with fine-grained details.

These results highlight the advantage of leveraging strong VLM priors in visuomotor frameworks, which can facilitate generalization and downstream reasoning, even when VQA is not a primary training objective.

## D  FAILURE CASE ANALYSIS

Despite FDVLA's robust performance in real-world tasks, we observed several types of failures in our experiments.

**1) Imprecise Manipulation under Ambiguous Perception.** When objects are visually ambiguous or partially occluded, such as in Montessori stacking with similar rings, FDVLA sometimes fails to detect boundaries or positions accurately. Poor lighting or shadows can make this worse, leading to grasp or placement errors. **2) Lack of Fine-Grained Force or Contact Sensing.** FDVLA only uses vision. In tasks that need precise force or contact, like assembling tight cubes or inserting blocks with friction, the robot can push too hard or fail to complete the insertion. The absence of tactile feedback makes these situations hard to handle. **3) Exceeding Robot Limits.** For complex or dual-arm tasks like stool assembly, the model occasionally gives actions that go beyond the robot's physical limits. This might include unreachable positions or movements that cause joint collisions. **4) Incomplete Dual-Arm Coordination.** When using both arms, FDVLA does not always synchronize them well. If one arm moves the object before the other finishes its part, objects can drop or assemblies may not finish. These problems are more likely when both arms must work together at the same time.

These failures show where FDVLA can improve. Adding other sensors like force or touch, making vision more robust to poor conditions, and improving dual-arm coordination are all promising directions for future work.

## E  LIMITATION AND FUTURE WORK

**Limitations.**  While FDVLA demonstrates strong visual reasoning and action generation across diverse manipulation tasks, several limitations remain. First, the current system does not incorporate force or tactile sensing, which is critical for reliably executing many real-world tasks involving physical contact and compliance. For example, during stool assembly, force feedback is essential for detecting successful insertion, adjusting alignment, and avoiding excessive force that could damage components. Human operators naturally rely on these capabilities, but they are currently unavailable in our purely vision-based framework. Second, our evaluations are conducted primarily in structured tabletop settings; generalization to unstructured or highly dynamic environments remains an open challenge. Third, FDVLA's reasoning and recognition abilities are inherently limited by the

coverage of the underlying pretrained vision-language model, especially for rare or domain-specific objects and instructions. Future work will explore the integration of additional sensing modalities and broader task diversity to address these limitations.

**Future Work.** Our FDVLA framework points to many directions for future work. Adding other types of sensors, like force, touch, or proprioception, could help the robot handle contact-rich or delicate tasks. Making FDVLA work well in unstructured or fast-changing environments is also important. This means dealing with new objects, occlusions, or changes in lighting. We are interested in trying larger-scale pretraining and lifelong learning. These could help the model pick up new skills over time and adapt to new situations. Finally, bringing humans into the loop, or making the system easier to teach, could make robots more helpful for people who are not experts.

## F  GENERALIZATION UNDER VISUAL AND SPATIAL VARIATIONS

### F.1  GENERALIZATION

We test FDVLA's robustness in four generalization scenarios using the same bowl placing task. As shown in Figure 7, each variant introduces a distinct real-world challenge. The results in Table 9 show that while performance drops under unseen conditions, FDVLA maintains solid success rates ($\geq 65\%$), especially under background and lighting variations. These findings suggest our system can generalize to modest environmental changes without retraining.

- **Unseen Object:** Replacing the bowl with a black baseball cap.
- **Unseen Spatial Position:** Elevating the bowl with a thick black mat.
- **Unseen Background:** Using a colorful map as the tabletop background.
- **Unseen Lighting:** Introducing strong direct lighting to produce reflections and shadows.

As shown in Figure 7, FDVLA completes the task successfully across all settings without any fine-tuning, indicating strong real-world robustness.

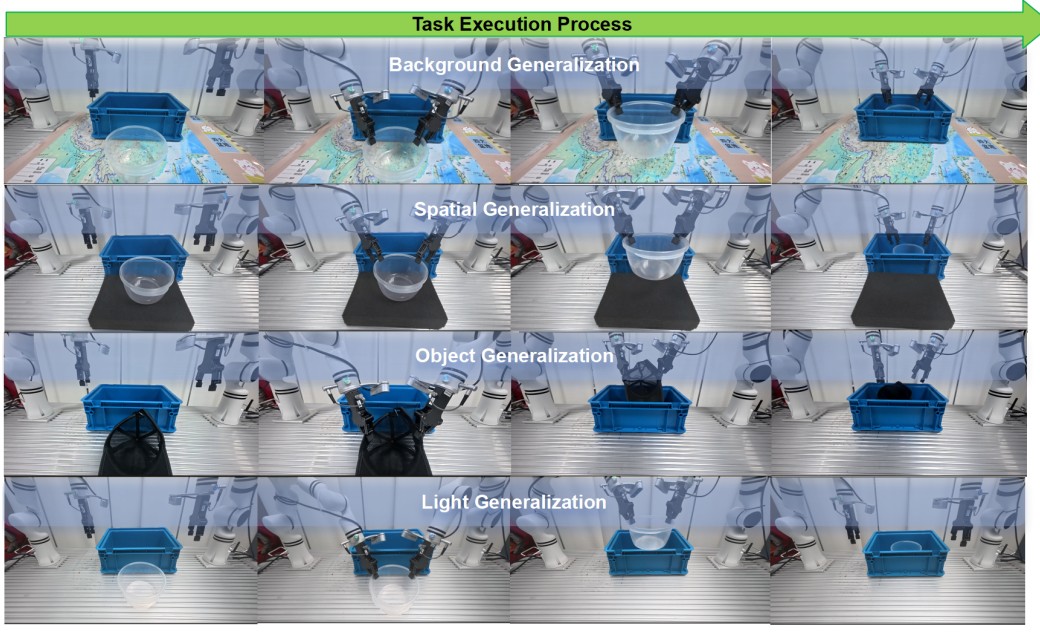

Figure 7: Generalization scenarios under different variations. FDVLA reliably executes the task with unseen object (black cap), changed spatial position (elevated bowl), cluttered background (map), and lighting perturbation (strong overhead light).

Table 9: FDVLA performance under generalization settings. Task success rate is reported as average over 20 rollouts per setting.

| Scenario | Success Rate | Drop (%) |
|---|---|---|
| Original | 0.80 | – |
| Unseen Object | 0.70 | -10% |
| Unseen Background | 0.75 | -5% |
| Unseen Position | 0.65 | -15% |
| Unseen Lighting | 0.70 | -10% |

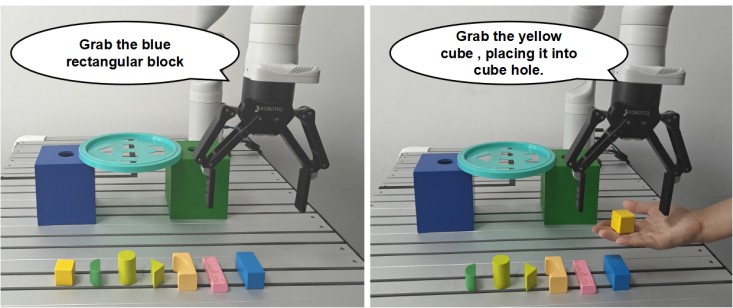

Figure 8: Reasoning-driven task reallocation. The model is instructed to "grab the blue rectangular block." Mid-execution, a yellow block is introduced by hand. Instead of ignoring it, FDVLA adapts its plan, reasoning that the yellow cube should be placed first. This flexible behavior reflects the model's ability to reinterpret a robot's actions.

### F.2 VARIATION: FLEXIBLE BEHAVIOR DRIVEN BY INTERNAL REASONING.

A key insight of FDVLA is its ability to generate reasoning phrases alongside action outputs, facilitating transparent interpretation of robot's actions . Figure 8 demonstrates how FDVLA adapts to unexpected input during execution. Initially tasked with "grabbing the blue rectangular block," the model redirects its action plan when a yellow block is introduced. Instead of resuming the original action, it decides to first pick and place the yellow block to its correct slot. This shows FDVLA's capacity to perform context-aware reallocation, driven by internal reasoning and dynamic scene understanding.

## G OTHERS

**Problem Statement.** At each timestep $t$, the robot observes multi-view RGB images $\mathbf{o}_t$, a natural language instruction $\mathbf{l}$, and its internal state $\mathbf{r}_t$ (e.g., joint angles). The goal is to predict the next-step action $\mathbf{a}_{t+1}$:

$$\pi : (\mathbf{o}_t, \mathbf{l}, \mathbf{r}_t) \to \mathbf{a}_{t+1} \tag{8}$$

Each action $\mathbf{a}_t$ encodes a 6-DOF end-effector pose and a 1-DOF gripper state. In the single-arm setting:

$$\mathbf{a}_t = [\Delta x, \Delta y, \Delta z, \text{ Roll}, \text{Pitch}, \text{Yaw}, \ g] \tag{9}$$

where $g \in \{0, 1\}$ denotes the force-adaptive gripper state (open or closed).

For dual-arm control, we concatenate the left and right arm actions to form a 14-DOF vector:

$$\mathbf{a}_t = [\mathbf{a}_t^{\text{left}}; \ \mathbf{a}_t^{\text{right}}] \in \mathbb{R}^{14} \tag{10}$$

We propose a unified *flow-diffusion* policy $\pi_{FDVLA}$ that generates actions by:

- predicting a velocity field $\mathbf{v}_t$ for coarse trajectory guidance,
- refining it with a residual $\epsilon_t$ via conditional denoising.

