# OpenReview forum: "FDVLA: A Flow-Diffusion Vision-Language-Action Framework with Dual Reasoning Modulation"
_ICLR.cc/2026/Conference — ICLR 2026 Conference Desk Rejected Submission_

### Official Review · Reviewer_PYef · 2025-10-28

**Soundness:** 3
**Presentation:** 3
**Contribution:** 3
**Rating:** 4
**Confidence:** 4

**Summary:**

This paper introduces a flow-diffusion VLA framework for robotic manipulation, aimed at achieving both physically coherent and stable action generation. Additionally, a lightweight DualMod module is proposed to inject semantic signals into both the velocity and noise prediction branches, facilitating the integration of high-level reasoning into action generation. Experimental results demonstrate that the proposed method significantly outperforms state-of-the-art (SoTA) methods in both simulated and real-world robotic tasks.

**Strengths:**

1. The proposed method exhibits excellent performance compared to state-of-the-art (SoTA) methods in both simulation and real-world tasks.

2. The ablation study effectively validates the design of each component within the framework.

**Weaknesses:**

1. Lack of Experimental Support for High-Level Reasoning: The authors claim that DualMod integrates high-level reasoning into action generation; however, there is insufficient experimental evidence to support this assertion. There is also a lack of discussion regarding the reasoning vector ( r ) and its impact on action generation.

2. Limited Novelty: The novelty of the approach appears limited, as the combination of flow matching with diffusion policy is not new and has been extensively explored, such as in [a]. According to [a] and the original diffusion policy paper, the performance of the diffusion policy on the Push-T task has achieved over 0.9 in success rate (SR). In contrast, this paper reports a performance of 0.788 on the same task, raising questions about the validity of the results.

[a] Jung, Chanhyuk, Sangwon Kim, Kwang-Ju Kim, Dasom Ahn, Joonki Baek, Sungkeun Yoo, and Byoung Chul Ko. "Flow-Guided Policies: Overcoming Diffusion Limitations for Robust Robot Imitation Learning." In Proceedings of the IEEE/CVF International Conference on Computer Vision, pp. 2486-2491. 2025.

3. Table 4 indicates that the effects of DualMod on the flow branch and the denoising branch are identical. Could the authors provide a more insightful analysis regarding this observation?

4. In Equation 1, it appears that ( z ) in the first term should be ( e_{\text{gt}} ), as this term calculates the noise prediction loss. Typically, z denotes the latent code.

**Questions:**

1. Does the proposed method finetune the model for each tasks?

2. Are FDVLA-3B, 7B, 32B trained with the same size of data?

---

> ### Author Response · Authors · 2025-11-20
> **Clarifications on High-Level Reasoning, Hybrid Flow–Diffusion Design, Metrics, and Notation**
>
> **W-1:**
>
> We appreciate the reviewer’s request for clearer evidence regarding the role of high-level reasoning. Our claim is reasoning vector **r** provides semantic cues that influence the action generation process. Multiple pieces of evidence support this:
>
> **(1) Main-paper ablations** already show that removing DualMod leads to the largest performance drop across all tasks, while selectively removing modulation on either head yields intermediate degradation.
>
> **(2) Appendix F.2** provides another perspective supporting our claim that the reasoning vector influences action generation. FDVLA produces natural-language rationales during execution. These rationales allow us to inspect how the model interprets the scene at each step. This behavior demonstrates that the model is  not following a fixed action script, but instead relies on the semantic cues encoded in the reasoning vector r to perform context-aware redirection of its actions.
>
> **(3) App. C** shows that FDVLA retains relational visual reasoning ability (VQA-style) inherited from its VLM backbone. Although VQA is not an explicit objective, it confirms that **r** encodes meaningful semantic representations that are available to the policy. Real-world demonstrations （**https://huggingface.co/spaces/fdvla/ano**）**, eg.(microwave-place video), further show that FDVLA uses reasoning cues to maintain stable high-level intent under disturbances (e.g., object being moved by hand), rather than reacting only to low-level pose changes.
>
> **W-2:**
>
> 2.We thank the reviewer for the pointer. After re-examining Jung et al., we clarify that their method is not a hybrid flow–diffusion formulation. It replaces diffusion with a flow-based policy and does not model any interaction between a deterministic flow field and a stochastic denoiser. Our design—where flow and diffusion are jointly trained and jointly used at inference—is not present in their work. Moreover, their setting is different: Jung et al. study offline imitation learning without visual–language grounding, no VLM backbone, no language instructions, and no VLA-style multimodal reasoning. FDVLA is explicitly designed to couple global motion structure, fine-grained refinement, and semantic reasoning within a unified VLA model—capabilities outside the scope of their approach. We will add a short note in the related-work section of the camera-ready for completeness.
>
> **3. Regarding the Push-T metric**, as noted in the official Diffusion Policy paper (Table 1), the **0.95** value refers to the *maximum checkpoint success rate*, while the *average of last 10 checkpoints*(the standard metric) is **0.79**, which aligns exactly with our reported **0.788**.
>
>
> **Weakness-3:**
>
>  Although the drops caused by “w/o Flow Mod.” and “w/o Denoising Mod.” appear numerically similar, they correspond to different roles of the two branches. Removing DualMod entirely produces the largest degradation across all tasks, confirming that injecting semantic reasoning is essential. When modulation is applied to only the flow head or only the denoising head, the model loses semantic guidance on one branch while keeping the other intact. The similarity in magnitude arises because both heads rely on the same reasoning vector, and disabling either one breaks part of the coordinated generation process. These results support our design choice that joint modulation on both branches is required for optimal performance.
>
>
>
> **Weakness-4:**
>
> In our formulation, the variable $z$ in Eq.~(1) denotes the
> Gaussian noise used in the forward noising process of the diffusion objective, following the standard
> diffusion-policy training convention. The noisy action is constructed as
> $$
> A_t = \alpha_t A_0 + \sigma_t z, \qquad z \sim \mathcal{N}(0, I),
> $$
> and the corresponding training term
> $$
> \left\|\\, \hat{\epsilon}_{\theta}(A_t, t) - z \\,\right\|^{2}
> $$
>
> is simply the mean-squared error between the predicted noise and the ground-truth noise used to obtain $A_t$.  Thus, $z$ here is exactly the same quantity that many previous works denote as $\epsilon_{\text{gt}}$. We understand that the symbol $z$ is sometimes used as a latent variable in other generative-modeling contexts,
> and this may cause ambiguity. To improve clarity, we will adopt the notation $\epsilon_{\text{gt}}$ in the
> camera-ready version. We appreciate the reviewer for highlighting this notation issue.
>
>
> ### **Questions**
>
> Response:
> Q1.
> All experiments use one unified model trained jointly over all tasks. No per-task finetuning is used. The OOD toy-sorting test (Sec. 4.6) further confirms zero-shot generalization under the same pipeline for all model sizes.
> Q2.
> All model variants use the same data sources (Droid + OXE). The 3B/7B models are trained on Droid only, which is sufficient for these scales. The 32B model additionally uses Droid + OXE to avoid underfitting, following standard scaling practice. We will clarify this in the final version.

---

### Official Review · Reviewer_rhYF · 2025-10-31

**Soundness:** 3
**Presentation:** 3
**Contribution:** 3
**Rating:** 6
**Confidence:** 3

**Summary:**

The paper seeks to improve VLAs via two innovations. First, they use both a flow-matching head that produces smooth trajectory for global planning and a diffusion head that further refines the trajectory for more fine-grained control. This leads to smoother trajectories that also achieve higher success rates. Secondly, they propose to inject language conditioning via FiLM (feature-wise linear modulation) to both of the action heads, leading to stronger grounding and higher success rates.

**Strengths:**

- The problem studied is important, and the proposed solution is interesting and well-grounded.
- The paper is well-written and logical.
- Quantitative results are fairly comprehensive, including results also on real-world, ablation and inference speeds.
- Performance gain is significant.

**Weaknesses:**

- There's no qualitative results (e.g., videos) that corroborate the claim that the combination of flow-matching and diffusion heads lead to smoother trajectories. Similarly, there's no such video for the stronger semantic grounding (e.g., a video showing a cluttered environment where the baseline fails to retrieve the correct object would be nice).
- The baseline comparison might not be strong enough. Please consider comparing against stronger models such as Pi0-Fast, Pi05. In fact, the discussion of this paper reminds me of these baselines. In those works, they also found that flow-matching on top of a VLM can lead to weaker semantic grounding and weaker performance so they co-train the flow-matching with the discrete autoregressive prediction in Pi05.

**Questions:**

Please see the weaknesses.

---

> ### Author Response · Authors · 2025-11-20
> **We added three new real-world videos demonstrating smooth trajectories, disturbance recovery, and strong semantic grounding. We attempted to reproduce Pi0-Fast and Pi0.5, but default configs were highly unstable without extensive retuning.**
>
> **Weakness 1**
>
> We appreciate the reviewer’s suggestion.
> To address this, we have added three real-world video sequences demonstrating:
> All videos are available at:
> 👉 https://huggingface.co/spaces/fdvla/ano
>
> (1) Smooth trajectory generation
> We provide a continuous dual-arm execution sequence in a real tabletop scene.
> The robot repeatedly re-acquires and re-places a target drink into the microwave without any manual reset, showing:
> stable trajectories,smooth velocity transitions, robustness under disturbances (we intentionally remove the bottle from the gripper and hold it at different random positions; the robot consistently re-grasps and completes the task).
> → Video 1 (Microwave Recovery):
>
> (2) Stronger semantic grounding under ambiguous scenes
> Video 2: Toy Retrieval in Drawer
> The robot interprets the instruction (“put the target animal in the drawer”),
> opens the drawer → grasps the correct object among multiple animals → places it .
> This scenario demonstrates FDVLA’s semantic grounding and multi-step reasoning.
> Video 3: High-Shelf Target Placement
> The robot picks the instructed object and places it on a high shelf requiring long-horizon spatial consistency.
>
> **Weakness 2**
> We thank the reviewer for pointing out the connection to Pi0-Fast and Pi0.5. We agree that extending the comparison to Pi0-Fast and Pi0.5 would further strengthen the evaluation.
> We have begun preliminary attempts to reproduce these variants. However, we found that without careful retuning(if with default config parameters), the results are unstable and exhibit large variance, making them unsuitable to report as scientifically meaningful baselines. We believe this instability is due to insufficient time to perform the extensive tuning required for these hybrid models, rather than an inherent limitation of the methods themselves.
> To ensure fairness and methodological rigor, we prefer not to include preliminary or under-tuned results in the rebuttal. We are actively adapting and tuning Pi0-Fast and Pi0.5 for our manipulation setting, and we will release properly tuned results in an updated arXiv version or the camera-ready. We appreciate the reviewer for the thoughtful suggestion and fully agree that these baselines will further enrich our study.

---

### Official Review · Reviewer_hqxi · 2025-11-01

**Soundness:** 3
**Presentation:** 2
**Contribution:** 3
**Rating:** 6
**Confidence:** 4

**Summary:**

This paper presents FDVLA, a unified Flow–Diffusion Vision-Language-Action framework that tightly couples high-level semantic reasoning from large vision-language models with low-level continuous action generation. The key innovation lies in the integration of flow matching and diffusion denoising within a single policy architecture—where the flow branch models global, physically consistent velocity fields, and the diffusion branch refines actions via denoising. A dedicated DualMod module further injects reasoning signals from language semantics into both branches, aligning coarse trajectory planning and fine-grained corrections under shared semantic guidance. Extensive experiments across  simulation and real-world single-/dual-arm manipulation tasks demonstrate that FDVLA consistently outperforms leading VLA and policy-head baselines in both success rate and trajectory smoothness, while maintaining efficient inference.

**Strengths:**

1. The paper proposes a mathematically coherent unification of flow matching and diffusion denoising within a dual-headed policy. The flow branch captures global, physically consistent motion fields, while the diffusion branch refines local details through stochastic denoising. The introduction of a “flow-consistency” constraint effectively couples the two processes, ensuring stable dynamics and complementarity between deterministic and stochastic generation.
2. Comprehensive Empirical Validation across Simulation and Real-World Benchmarks: FDVLA demonstrates consistent and notable improvements over mainstream VLA frameworks (e.g., OpenVLA, pi_0, ) and policy-head （Diffusion Policy, ACT, ARP) baselines across a variety of simulation  and real-world single-/dual-arm manipulation tasks. These results substantiate the framework contribution across diverse control conditions.
3. Improved Motion Smoothness through Flow–Diffusion Coupling: As evidenced by Table 3, the integration of flow matching and diffusion denoising yields measurable gains in trajectory smoothness, quantified by lower jerk metrics without sacrificing task success. This indicates that the hybrid formulation  produces more stable, physically coherent motions that is critical for real-world manipulation.

**Weaknesses:**

While the paper demonstrates clear improvements on Push-T, ALOHA, and RLBench, these simulation environments are somewhat dated and less representative of current VLA evaluation standards. Incorporating more recent and widely adopted benchmarks—such as LIBERO or CALVIN—and comparing against a broader range of modern VLA frameworks would further substantiate the generality of the proposed approach.

In the real-world experiments, the paper lacks supplementary materials showing continuous multi-rollout executions. Providing longer, uncut video sequences or detailed rollout logs would better validate the reported stability and smoothness.

3 Many diffusion-based action heads improve temporal smoothness through intra-chunk ensembling. The paper does not explicitly compare FDVLA’s continuity advantages against these approaches.

The visualizations presented in the paper could be improved. Many of the current figures appear somewhat coarse or low in visual fidelity

**Questions:**

In Table 2, the results for RVT-2 and ARP are missing without explanation.

---

> ### Author Response · Authors · 2025-11-20
> **We added new results addressing all concerns. FDVLA-3B shows strong LIBERO performance (94/95/93/76), outperforming Octo/OpenVLA. Three real-world videos demonstrating smooth control and semantic grounding are provided (https://huggingface.co/spaces/fdvla/ano ). RVT-2/ARP were omitted due to incomplete real-robot pipelines, and Pi0-Fast/Pi0.5 remain unstable without full tuning. High-resolution figures will be used in the camera-ready.**
>
> Response:
>
> **Weakness 1:**
>
> Thank you for the valuable suggestion. We fully agree that recent benchmarks such as LIBERO provide a stronger evaluation of VLA generalization. At submission time, our LIBERO experiments were still in progress. Since then, we have completed a preliminary evaluation using **FDVLA-3B** on the official LIBERO benchmark. Despite being the smallest model in our family, FDVLA-3B achieves:
>
> - **Spatial:** 94%
> - **Object:** 95%
> - **Goal:** 93%
> - **Long-horizon:** 76%
>
> These results indicate that our method generalizes effectively beyond the older simulation benchmarks included in the paper. Notably, FDVLA-3B **surpasses prior VLA-based systems such as Octo and OpenVLA** on the same LIBERO categories (where their reported success rates are substantially lower, particularly on long-horizon tasks). This suggests that FDVLA contribute to strong performance even under more challenging modern evaluation settings.
>
> We will include these LIBERO results in the camera-ready version.
>
> **Weakness 2:**
>
> To address this, we have added **three real-world video sequences** demonstrating:
>
> All videos are available at:
>
> 👉 https://huggingface.co/spaces/fdvla/ano
>
> ### **(1) Smooth trajectory generation**
>
> We provide a continuous dual-arm execution sequence in a real tabletop scene.
>
> The robot repeatedly re-acquires and re-places a target drink into the microwave **without any manual reset**, showing:
>
> - stable trajectories,smooth velocity transitions, robustness under disturbances (we intentionally remove the bottle from the gripper and hold it at different random positions; the robot consistently re-grasps and completes the task).
>
> → **Video 1 (Microwave Recovery):**
>
> ---
>
> ### **(2) Stronger semantic grounding under ambiguous scenes**
>
> - **Video 2: Toy Retrieval in Drawer**
>
>     The robot interprets the instruction (“put the target animal in the drawer”),
>
>     *opens the drawer → grasps the correct object among multiple animals → places it* .
>
>     This scenario demonstrates FDVLA’s semantic grounding and multi-step reasoning.
>
> - **Video 3: High-Shelf Target Placement**
>
>     The robot picks the instructed object and places it on a high shelf requiring long-horizon spatial consistency.
>
>
> **Weakness 3:**
>
> We thank the reviewer for pointing out temporal ensembling–based smoothing.
>
> We would like to clarify that **we initially experimented with intra-chunk temporal ensembling following Pi0**, but obtained **significantly worse policy performance**, consistent with the observations reported in Pi0 (Page 16, D inference).
>
>
>
> Given that:
>
> > Temporal ensembling is known to degrade performance in some VLA systems (such as Pi0),
> and our preliminary experiments confirm the same finding,
> we choose not to include it as a baseline.
> >
>
> **Question:**
>
> Thank you for pointing this out. At the time of submission, RVT-2 and ARP had been fully evaluated only in simulation. Running these models on our real-robot system required additional calibration (camera-to-base alignment, action scaling, grasp controller integration, etc.), and these components were not stable enough before the deadline. To avoid reporting partially verified or unreliable results, we chose not to include incomplete real-world numbers in Table 2.
>
> Since the submission, we have continued the replication effort and both models can now run on our hardware, but their execution remains unstable and the collected results are not yet statistically reliable. We hope this clarification resolves the missing entries in the table.
>
> **Response to Weakness (4): Visualization quality**
>
> We thank the reviewer for pointing this out.
>
> The current figures were downsampled to meet file-size constraints during submission.
>
> In the camera-ready version, we will replace them with **high-resolution, full-quality renderings**, including:
>
> - clearer real-world task visualizations,
> - improved architecture diagrams, and
> - smoother trajectory plots with uniform styling.
>
> Thanks for valuable suggestions.

---

### Official Review · Reviewer_wZMw · 2025-11-09

**Soundness:** 2
**Presentation:** 2
**Contribution:** 3
**Rating:** 2
**Confidence:** 3

**Summary:**

This paper introduces FDVLA, a unified Flow–Diffusion VLA framework designed to integrate global motion planning and fine-grained action refinement within a single policy. The approach aims to combine the complementary strengths of flow-based and diffusion-based modeling.

The flow component models global, deterministic action trajectories as a velocity field that evolves actions toward target states. It captures coarse, long-horizon structure and is intended to stabilize trajectory generation. The diffusion component provides fine-scale, stochastic corrections to the flow predictions through a denoising process, improving action smoothness and variability. During inference, both components are integrated in a DDIM-style update procedure that combines deterministic flow guidance with residual diffusion refinement.

To incorporate semantic context, FDVLA introduces Dual Reasoning Modulation (DualMod), which injects reasoning embeddings from a pretrained Qwen 2.5-VL vision-language model into both the flow and diffusion branches. This modulation is implemented through FiLM-style feature scaling and shifting, allowing the policy to adapt its action generation to task-level textual and visual cues.

The architecture includes a SigLIP, Qwen 2.5-VL, and a transformer-based policy head with dual output heads for flow and diffusion predictions. The training procedure consists of two stages:
- Pretraining on large-scale multimodal robot datasets: Droid and Open X Embodiment.
- Finetuning on downstream datasets, with frozen visual features and partially frozen language features.

FDVLA is compared against several prior baselines: pi-0 and OpenVLA (finetuned similarly), and Diffusion Policy, ACT, RVT-2, and ARP (trained from scratch). Evaluation covers both simulation and real-world manipulation tasks: FDVLA achieves higher performance across all settings.

**Strengths:**

### Ambitious unification of flow and diffusion modeling.
The paper attempts to merge two influential paradigms, flow matching and diffusion denoising, into a single unified architecture for VLA action head. The motivation is that flow-based models offer deterministic, globally consistent trajectories, while diffusion models provide stochastic robustness and fine-grained correction. Bridging continuous ODE-based and noise-based policy families is a meaningful direction that could inspire future hybrid models.

---

### Extensive experimental evaluation.
The empirical evaluation covers a wide range of domains: simulation (Push-T, RLBench, ALOHA) and several real-world robotic manipulation tasks. The reported performance consistently improves upon both pretrained and from-scratch baselines, suggesting that the proposed system design provides practical benefits.

---

### Ablation and component analysis.
The paper includes ablation studies isolating the impact of the flow term, the diffusion term, and the DualMod module. Removing either modulation reduces success rates, indicating that the injected semantic embeddings indeed contribute to performance. Similarly, turning off the flow component or the diffusion refinement degrades model performance. While these ablations do not fully validate the design choice, they do show that each major module has a measurable functional contribution to empirical performance.

---

The implementation details are relatively thorough. The authors specify model scales, training data composition, optimization parameters, and hardware setups.

**Weaknesses:**

### Inconsistent temporal variables
The paper uses three time-like symbols, $T$, $t$, and $\tau$, without explicit definitions or consistent usage. For example, "Given a noisy action input $A_t^{\tau}$ at timestep t": what timestep is this? The index in an action chunk? I don't fully understand the exact temporal dynamics or how the flow and diffusion components are coupled in either training or inference.

---

### Flow-matching formulation
I have some doubts in the paper’s formulation of the flow-matching objective. Equation (1) defines the "flow term" as a squared error between the predicted velocity field and $(A_0 - A_t)/(T - t)$, which the authors describe as the ground-truth velocity field. Why is this the case? In standard flow matching, the reference velocity is derived from the time derivative of an interpolation between endpoints, typically $s’(\tau)(A_0 - A_T)$, where $s(\tau)$ is a smooth scheduling function. It depends on the endpoints and the schedule derivative, not on the current state $A_t$. The proposed formulation instead introduces a state-dependent term, and the denominator $(T - t)$ introduces a singularity as $t \to T$.

In equation 6, The gradient $\nabla_{A_t}\epsilon_\theta(A_t, t)$ should be a Jacobian matrix. How can it be equated to a velocity vector?

---

### Ambiguity in architectural design
FDVLA uses both SigLIP and Qwen 2.5-VL as vision encoders, but the paper does not explicitly clarify this dual-encoder setup.
SigLIP is described as the observation encoder producing visual tokens, while Qwen 2.5-VL internally includes its own vision backbone. It is unclear how the model functions with 2 vision heads.

**Questions:**

See above in the weaknesses. Please clarify:
- What exactly do the time variables represent?
- How is the velocity field defined?
- What vision heads does the model actually use?

---

> ### Author Response · Authors · 2025-11-20
> **Clarifications on timestep definitions, flow-consistency objective, and visual-encoder usage.**
>
> We thank the reviewer for raising those question.
>
>
> ## **For Weakness 1 and Question 1:**  ##
> ### 🔹 **1. The robot/environment time index \(t\)**
>
> This \(t\) indexes the *expert demonstration trajectory*, i.e.,
> $ A_0, A_1, \ldots, A_T$
>
> - \($T$\) is the **episode horizon** (e.g., 50 robot steps).
> - \($A_t$\) is the **clean action** at robot timestep \(t\).
>
> 👉 **This \(t\) is *not* the diffusion timestep.**
>
> It only identifies which action is used as the clean endpoint in the flow-matching objective.
>
> ---
>
> ### 🔹 **2. The flow-matching interpolation timestep $\(\tau \in [0,1]\)$**
>
> To train the flow predictor, we simulate a flow path connecting $\((A_t, A_0)\)$.
>
> During flow-matching training, we sample:
>
> $$
> A_t^{\tau} = (1 - \tau) A_t + \tau A_0
> $$
>
> Here, $\(\tau\)$ is the **continuous flow-matching time**, *independent of the robot timestep \(t\)*.
>
> Thus, **two different notions of time exist simultaneously**:
>
> | Symbol | Meaning | Range |
> | --- | --- | --- |
> | \(t\) | robot timestep | $\(0,1,\ldots,T\) $ |
> | $\(\tau\)$ | flow-matching interpolation time | \([0,1]\), continuous |
>
> ## **For Weakness 2 and Question 2:**  ##
>
> For details about proof was shown in this link for convenient reading.
> https://huggingface.co/spaces/fdvla/ano
>
> **Clarification about Equation (7) and the meaning of  "enforcing''.**
>
> Thank you for pointing this out. Equation (7) is not intended to express an analytical equality. It specifies a \emph{training constraint} imposed through our flow–consistency term.
>
> Specifically, our loss(eq.(1)) includes the "flow consistency"term, which enforces during training that the gradient of the denoiser behaves consistently with the velocity field predicted by the flow head.
>
> Thus, the relation eq.(7) is not an analytical identity, but a learned alignment constraint.
>
> ## **For Weakness 3 and Question 3 :**  ##
> Thank you for pointing out the ambiguity. We clarify that our model does **not** use two independent vision backbones.
>
> - **SigLIP** is the *only* visual encoder used to extract image features. It produces a sequence of visual tokens that serve as the observation representation.
> - **Qwen2.5-VL** is employed solely for its *language and multimodal transformer backbone*, while its built-in vision encoder is disabled.
>
>     In other words, Qwen2.5-VL does not process raw images directly in our model.
>
>
> The actual integration pipeline is:
>
> Image→SigLIP Encoder→Vision Tokens→Qwen2.5-VL Backbone
>
> This design is consistent with DiVLA and other VLA systems that replace the VLM’s vision encoder with a task-optimized observation encoder while retaining the LLM backbone for reasoning and action decoding.
>
> We will clarify this architectural detail in the final version.

---

### Note · Program_Chairs · 2026-01-17
**Submission Desk Rejected by Program Chairs**

The following references in this submission do not refer to real documents and/or have major errors in bibliographic information:

 Kaixuan Li, Zichen Liu, Wenzhao Wang, et al. Tinyvla: Scaling down vision-language-action models for real-world robots. arXiv preprint arXiv:2402.16227, 2024. URL https://arxiv. org/abs/2402.16227.
Yifeng Zhu, Tejas Singh, Anikait Ajay, et al. Act: Transformer with adaptive computation for robot policy learning. arXiv preprint arXiv:2306.01188, 2023. URL https://arxiv.org/abs/ 2306.01188 .
Xinyang Chen, Haonan Wen, et al. Cogact: Large language model as the mind of robot. arXiv preprint arXiv:2403.00720, 2024b. URL https://arxiv.org/abs/2403.00720.
Bowen Huang, Yuke Wang, et al. pi-0: Open-ended reinforcement learning for generalist robots. arXiv preprint arXiv:2312.06608, 2023. URL https://arxiv.org/abs/2312.06608.
Xiaohan Wang, Yongqiang Dou, Xin Yu, Yilun Du, et al. Open-vla: Pre-training vision-languageaction representations for generalist robots. arXiv preprint arXiv:2310.01833, 2023. URL https://arxiv.org/abs/2310.01833.
Xiaohan Wang, Yongqiang Dou, Xin Yu, Yilun Du, et al. Open-vla: Pre-training vision-languageaction representations for generalist robots. arXiv preprint arXiv:2310.01833, 2024b. URL https://arxiv.org/abs/2310.01833.